# Hemoglobin-Based Oxygen Carriers: Where Are We Now in 2023?

**DOI:** 10.3390/medicina59020396

**Published:** 2023-02-17

**Authors:** Lin Chen, Zeyong Yang, Henry Liu

**Affiliations:** 1Department of Anesthesiology, Hubei Women & Children’s Hospital, 745 Wuluo Road, Hongshan, Wuhan 430070, China; 2Department of Anesthesiology, International Peace Maternity and Child Health Hospital, Shanghai Jiaotong University School of Medicine, Shanghai Key Laboratory of Embryogenic Disease, Shanghai Municipal Key Clinical Specialty, 1961 Huashan Road, Shanghai 200030, China; 3Department of Anesthesiology & Critical Care, Perelman School of Medicine, The University of Pennsylvania, 3400 Spruce Street, Philadelphia, PA 19104, USA

**Keywords:** blood substitute, oxygen therapeutics, HemAssist, PolyHeme, Hemopure, hemoglobin, genetically engineered, OxyVita, Sanguinate

## Abstract

The pursuit for blood a substitute has spanned over a century, but a majority of the efforts have been disappointing. As of today, there is no widely accepted product used as an alternative to human blood in clinical settings with severe anemic condition(s). Blood substitutes are currently also termed oxygen therapeutics. There are two major categories of oxygen therapeutics, hemoglobin-based and perfluorocarbon-based products. In this article, we reviewed the most developed but failed products and products still in active clinical research in the category of hemoglobin-based oxygen carriers. Among all of the discussed hemoglobin-based oxygen therapeutics, HemAssist, PolyHeme, Hemolink, Hemospan, and Hemoximer were discontinued. Hemopure is in clinical use in South Africa and Russia. Oxyglobin, the sister product of Hemopure, has been approved for veterinary use in the European Union and the United States. HemO2life has recently been approved for organ preservation in organ transplantation in the European Union. OxyVita and Sanguinate are still undergoing active clinical studies. The field of oxygen therapeutics seems to be entering a phase of rapid growth in the coming 10–20 years.

## 1. Introduction

Blood is a critically important substance in maintaining human life. One of the key functions of blood is transporting oxygen from the lungs to all body tissues to meet their metabolic requirements [1,2]. There are numerous etiologies to cause blood loss in humans, which generates the need for blood replacement therapy. In the ancient times, medical practitioners attempted to use various materials such as plant resins, milk, beer, chicken or sheep blood, and even human urine as a replacement for human blood [3,4]. In modern medicine, the demands for allogeneic blood transfusions for patients in various potentially life-threatening conditions are tremendously high [3,5]; the shortage of human blood for transfusions has been at critical levels in many locations worldwide [5,6]. Clearly, we need an alternative to human blood, which can carry oxygen to our body tissues as normal blood does, or oxygen therapeutic agent(s), as currently termed. There are other issues related to human blood transfusions, such as special storage requirements and storage-related problems (contamination risk), risks of transmitting infectious diseases (HIV, hepatitis), hemolytic and immunogenic complications, limited shelf-life, and poor portability. All these present significant challenges for the utilization of allogeneic blood in military and civilian settings [5,6]. Normally, oxygen is carried by hemoglobin inside red blood cells (RBC). Can hemoglobin molecules carry oxygen and stay in the blood circulation without the protection of an RBC membrane? Dr. Amberson reported in 1933 that bovine hemolysates could transport oxygen in experimental animals [3,7]; later, he attempted using an acellular hemoglobin solution to carry oxygen as an alternative to human blood [8]. However, the early studies on cell-free hemoglobin solutions seemed to indicate that an acellular hemoglobin solution was not a viable alternative to human blood due to significant side effects, such as renal failure, cardiac toxicity, and hypertension. These discouraging results slowed the pace of seeking an alternative to allogenic human blood to some extent until the discovery of the human immunodeficiency virus (HIV). In the 1980′s, the emergence of HIV pandemic and prevalence of hepatitis C transmission from human blood transfusion re-ignited the strong interests in alternative to human blood. The pursuit for such an allogeneic blood alternative was significantly more active since the 1980s, though the research efforts had been ongoing for many decades previously [3,5]. Although no widely accepted or globally approved blood substitute exists, which has been very frustrating, several products are currently approved in certain countries for certain indications [9,10,11]. Hemopure, a hemoglobin-based oxygen carrier, has been approved for clinical applications in human beings in South Africa [5], while the FDA granted Hemopure expanded access in the United States. Perftoran, a perflurocarbon-based product, has been approved for clinical use in Russia and Mexico [10,11], and it is being rebranded as Vidaphor by Fluor02 Therapeutics, Inc., Boca Raton, FL, USA, hoping to achieve FDA approval in the future. Hemo2life, developed by Hemarina company in France, recently also achieved approval for clinical organ preservation in the European Union [9]. Though most research activities and products are here in the US, FDA is more reluctant to approve these oxygen carriers than other authorities. Two conceptual changes seem to be occurring in this field in recent years; one is that the term “oxygen therapeutic” is being used to replace “blood substitute”. This change reflects the reality that most current searches are for the replacement of RBC’s oxygen carrying function, and that instead of searching for a product comparable to RBC transfusion, researchers are seeking a product which will lower the mortality and morbidity in life-threatening anemic situations [5,12]. The clinical benefits of blood substitutes/oxygen therapeutics can potentially include storage at room temperature with prolonged shelf-life, no disease transmissions, universal compatibility without blood typing, no antigenic reactions, enhanced oxygen delivery, potential abundant supply, no immunologic effects, and potentially improved rheologic properties. There are basically two major categories of oxygen therapeutics: hemoglobin-based oxygen carrier (HBOC) and perflurocarbon-based oxygen carrier [3,5]. In this review article, we will focus on discussing HBOCs’ current development.

## 2. Hemoglobin-Based Oxygen Carriers (HBOCs)

### 2.1. Function of Red Blood Cells and Acellular Hemoglobin

The primary function of the blood is transporting oxygen (O_2_) from the lungs to the body tissues and carrying carbon dioxide from tissues to the lungs. RBCs in the blood achieve the oxygen-carrying function by its hemoglobin’s gas binding capability [1]. Hemoglobin is a tetrameric protein with two α- and two β-polypeptide chains; each chain has an iron-containing heme group, which is capable of binding one oxygen molecule. The oxygen dissociation curve displays a sigmoidal shape of the oxygen-binding equilibrium [13]. The oxygen-carrying iron in the hemoglobin molecule is normally in the reduced “ferrous” (Fe^2+^) state; it is unable to bind oxygen if the hemoglobin molecule is oxidized to the “ferric” state (Fe^3+^) to form methemoglobin (MetHb) [1,13]. The unique ability of hemoglobin molecules to undergo conformational changes enables the hemoglobin molecules to bind/load more oxygen in the lungs (higher oxygen affinity) and release more oxygen in the tissue capillaries (lower oxygen affinity). Some allosteric effector molecules, 2,3-diphosphoglycerate (2,3-DPG), which form inside RBCs as a glycolytic intermediate, helps hemoglobin’s reversible conformational regulation of oxygen-binding affinity [1,13]. Thus, in the design of oxygen-carrier molecules for the purpose of substituting allogeneic human blood, it is imperative to maintain such oxygen-carrying thermodynamic and kinetic characteristics of natural hemoglobin molecules, to maintain the redox metabolic environment, and to minimize/avoid the irreversible methemoglobin formation [14]. In the acellular hemoglobin solution, hemoglobin molecules lack the protective RBC cell membrane and biological environment. The hemoglobin tetramer can potentially be rapidly degraded into its dimeric and monomeric protein units, which will quickly be cleared from the intravascular blood circulation into the extravascular space and removed by the kidneys. Stroma-free hemoglobin is also devoid of oxygen affinity regulatory allosteric effectors such as 2,3-DPG, and has no protective anti-oxidant enzymes as well. Therefore, such hemoglobin molecules have a dysregulated tissue oxygenation capacity when compared with RBC-encapsulated hemoglobin, and they are also prone to rapid irreversible oxidation to methemoglobin, thus losing their oxygen transporting ability. Moreover, stroma-free hemoglobin is a potent scavenger of intravascular and extravascular nitric oxide produced by vascular endothelial cells as a natural potent vasodilator. This is believed to be the mechanism of the acellular hemoglobin’s hypertensive side effects. As such, a HBOC should provide efficient tissue oxygenation while maintaining reasonable intravascular circulation life-time, inducing minimal hypertensive side effects, and causing no/minimal free hemoglobin induced toxicity. These are the three hurdles any HBOC product needs to overcome, and these are the three prominent design requirements for new generations of HBOCs [14].

### 2.2. Mechanisms of Hemoglobin Molecular Modifications

#### 2.2.1. Polymerization

Polymerization of hemoglobin molecules can significantly increase the size of acellular hemoglobin, thus minimizing the extravasation and prolonging their half-life in intravascular circulation. Polymerization is mainly achieved with glutaraldehyde. Polymerized hemoglobin was considered to be the key factor in mitigating the vasoconstriction effects observed in the early HBOC studies. The first generation of polymerized HBOCs includes HemAssist, Hemolink, PolyHeme, and Hemopure (Oxyglobin, the veterinary version of Hemopure) [14].

#### 2.2.2. Cross-Linking

Hemoglobin molecules can be cross-linked between two α-chains or two β-chains. The linkers can be either diaspirin or raffinose. It is believed that α-α chains cross-linking can prevent dissociation of oxyhemoglobin into αβ dimers, which are usually readily excreted by the kidneys; the cross-linked tetramers stabilize the molecule, thus preventing renal filtration [6,14]. The intermolecular cross-linkage between the two α- and the two β-subunits using a site-specific crosslinker, diaspirin, helps to stabilize the hemoglobin molecules.

#### 2.2.3. Conjugation

Hemoglobin molecular conjugation with antioxidant enzymes can potentially protect hemoglobin molecules against free radicals. Conjugated antioxidant enzymes also increase the size of hemoglobin molecules, which renders them significantly less cleared by the kidneys and less penetrating into vascular walls, thus less nitric oxide scavenging.

In addition to the above-mentioned three basic mechanisms of hemoglobin molecular modification, any of the above two mechanisms can be combined to form more complemental products as cross-linked and polymerized, or cross-linked and conjugated hemoglobin, as shown in Table 1 [3,5,12].

#### 2.2.4. Encapsulation

The hemoglobin molecules from humans or animals can be encapsulated inside a phospholipid bilayer capsule, which will mimic the cellular membrane of RBCs. Dr. Thomas Chang et al. conducted the very first experiment by encapsulating free hemoglobin molecules as early as 1957 [15]. The phospholipid bilayer with cholesterol molecules functions as the RBC membrane, which provides increased rigidity and mechanic stability for liposome-encapsulated actin-hemoglobin [15]. Recently, a phase I clinical trial in humans was approved in Japan in November 2022 [16,17].

#### 2.2.5. Genetically Engineered Recombinant Hemoglobin

Genetically engineered hemoglobin production involves the application of recombinant DNA technologies. Basically, the modified human hemoglobin genes are introduced via plasmid into *E. Coli* or *yeast*. The modified human Hb genes are then expressed inside *E. Coli* to produce human hemoglobin molecules. Large quantities of recombinant human hemoglobin can also be produced and purified from transgenic *E. Coli*. Protein-engineering strategies can be applied to enhance the physiological suitability of these hemoglobin molecules to address specific oxygen-carrying efficacy and toxicity issues. By mutagenesis of hemoglobin molecules, we can also adjust dioxygen affinity, reduce nitric oxide scavenging, slow the auto-oxidation process, hamper subunit dissociation, and potentially minimize irreversible subunit denaturation [4,18]. This technique generally carries minimal concerns pertinent to transmitting infectious diseases.

### 2.3. Natural Hemoglobin

Human hemoglobin molecules are contained inside and protected by RBCs when these hemoglobin molecules transport oxygen to body tissues in blood circulation. However, hemoglobin molecules are not inside the RBCs in some invertebrates such as polychaete *Arenicola marina* and earthworms. These acellular hemoglobin molecules circulate in the blood and accomplish their oxygen-carrying function in these animals. Scientists are trying to investigate whether these acellular hemoglobin molecules can be utilized to carry oxygen in human blood. Hemoglobin molecules from earthworms are still in laboratory research and no established products have been developed yet [19,20]. Hemarina, a French company, developed HemO2Life from the natural hemoglobin of polychaete *Arenicola marina* [20]. HemO2life has achieved the European Union’s approval for human organ preservation.

## 3. Past and Present HBOC Products Approved for Clinical Trials

Here we will discuss some of the most representative products which either developed in the past and failed or are still in clinical trials.

### 3.1. HemAssist

The first human use of acellular hemoglobin was reported in 1949 by Dr. Amberson and his team; they used a hemoglobin solution in a parturient suffering from a massive hemorrhage after delivery [21]. The first modern oxygen carrier approved by the FDA for clinical trial was HemAssist, a product developed jointly by the US military and the Baxter Corporation. HemAssist was a tetrameric, 2,3-diaspirin cross-linked hemoglobin (α-α cross-linked) from expired human blood [22,23]. In the Phase I and II clinical trials, HemAssist was overall well tolerated by the patients enrolled. In addition, a blood sparing effect was also demonstrated in patients enrolled in the Phase III trial; however, these surgical patients enrolled appeared to have an increased risk of pancreatitis and myocardial infarction. Trauma patients enrolled in the US Phase III trial had significantly higher mortality in the treatment group, though the mortality seemed to be evenly balanced out in the European Phase III trial. HemAssist was terminated in 1999 [22].

### 3.2. PolyHeme

PolyHeme was a product developed by Northfield Laboratories (Evanston, IL, USA) after the Vietnam War from expired human blood. PolyHeme underwent all Phase I, II, and III clinical trials. It was successful in elevating the plasma hemoglobin level of patients in the treated group receiving PolyHeme instead of allogenic human blood. However, PolyHeme failed to meet the non-inferiority goal for 30-day mortality in Phase III clinical trial. The FDA denied its Biologic License Application for approval in May 2009, and Northfield Laboratories filed for bankruptcy. That indicated the end of PolyHeme’s journey to clinical use [24].

### 3.3. HemoLink

Hemolink was developed by Hemosol Inc. (Mississauga, ON, Canada). Hemolink was a unique hemoglobin solution containing stabilized tetrameric human hemoglobin A units, which were cross-linked, and o-raffinose polyaldehyde polymers [25]. The product showed some potential in the Phase III clinical trials and saved some patients’ lives. However, Hemolink did not achieve marketing authorization in Canada. Hemosol halted its clinical trials in 2003 and declared bankruptcy in 2005 [5].

### 3.4. Hemopure

Hemopure (also known as Hb-201) was developed by Biopure corporation (Cambridge, MA, USA), and it is a chemically stabilized, cross-linked bovine hemoglobin dissolved in a salt solution intended for human use. Biopure also developed the same product with the trade name Oxyglobin, which was approved for veterinary use in dogs in the US in 1997 and in Europe in 1998. Hemopure as an alternative to allogeneic human blood was approved in South Africa in 2001 and in Russia in 2006, respectively [26]. It also achieved FDA expanded use approval in the United States. Hemopure was evaluated in human cardiac trials, in which it showed increased coronary oxygenation and perfusion. Hemopure is unique because it does not need pulsatile blood flow to reach ischemic tissues due to its small polymer size and potent ability to upload oxygen in the lungs and offload oxygen at the tissue level. Hemopure seemed to be safe in patients under 80 years old who did not have severe pre-existing illness. Unfortunately, Biopure Corporation was unable to obtain FDA approval for full clinical use, filed for bankruptcy protection in 2009, sold its assets to OPK BioTech (Cambridge, MA, USA) in 2009, and later sold to Hemoglobin-O_2_ Therapeutics (Souderton, PA, USA) in 2014 [26]. Thereafter, the new owner, Hemoglobin-O_2_ Therapeutics, continued its efforts in promoting Hemopure. One of their recently sponsored studies found that Hemopure could potentially restore brain circulation and cellular functions up to four hours after death in animal models. The finding was published in the April 2019 issue of *Nature*. A remarkable reduction in cell death and restoration of some cellular cerebral functions, including spontaneous synaptic activity, were reported. In this study, Hemopure was an essential ingredient in transferring and mobilizing oxygen to the brain. Studies investigating the dynamics of cellular recovery following prolonged global ischemia further validated recent preclinical and clinical data that demonstrated the ability of Hemopure to perfuse organs such as the kidney and liver, showing that Hemopure could potentially replace packed red blood cells in organ transplantation [26].

### 3.5. HemO2life

HemO2life was developed by Hemarina company in France. It was derived from a natural extracellular hemoglobin isolated from the polychaete *Arenicola marina*. The unique features of HemO2life make it a potential new therapeutic vehicle in providing tissue oxygenation for various clinical situations [20]. HemO2life is also indicated for donor organ preservation before transplantation as a tool of preserving donor organ function in preclinical and clinical settings for which it is currently approved in the European Union. HemO2life is proven to be safe as an additive to organ preservation solutions and has a beneficial effect on ischemia/reperfusion injuries. HemO2life has numerous potential advantages and indications in clinical settings and other scenarios [20]. The polychaete *Arenicola marina* is a very old species, living on earth since 450 million years ago, and it colonizes the intertidal area of the east-Atlantic shoreline of France from the North Sea to Biarritz. HemO2life has some unique features: (1) a naturally extracellular and polymerized hemoglobin with a molecular weight about 50 times that of human Hb; (2) functional O_2_-binding and -liberating properties similar to human Hb; (3) the ability to bind 156 molecules of oxygen vs 4 in the case of human Hb; (4) naturally antioxidative properties due to an intrinsic superoxide-dismutase-like activity. Hopefully in the near future, HemO2life will obtain the approval to be used for anemic situations in the European Union and other countries. HemO2life was tested in COVID-19 patients, and it was found that it could increase the arterial O_2_ content in a situation when pulmonary exchange failed, as O_2_ binding and release occur passively in a simple O_2_ gradient. HemO2life could improve COVID-19 patients’ survival, as it minimized tracheal intubation, shortened oxygen supplementation, and offered the possibility of treating many patients when ventilators were lacking [27].

### 3.6. Hemoglobin Vesicles

Hemoglobin vesicles (HbV) are cellular-structured HBOCs encapsulating a purified and concentrated hemoglobin solution in pegylated phospholipid vesicles. They have a liposome configuration with a mean particle diameter of 225–285 nm, which shields all potential toxic effects of hemoglobin molecules by the artificial lipid bilayer membrane and by mimicking the erythrocyte cell membrane [28]. HbV was described as a resuscitative fluid extensively in animal models, plus pharmacokinetics, biodistribution, excretion, immunological response, and hematological studies. The first-in-human phase I clinical trial from 2020 to assess HbV’s safety and pharmacokinetics in healthy male adult volunteers found that HbV is safe and efficacious. HbV can potentially be used as an alternative to allogeneic blood transfusion as well as for oxygen and carbon monoxide therapy, perfusate for transplant organs, and photosensitizer [15,28,29]. Permission for a Phase I clinical trial was also granted to Dr. Sakai’s group in Japan [28].

The World Health Organization believes that synthetic biology and the development of engineering techniques have provided unique opportunities for global scientists to make promising hemoglobin production [29]. The various diverse and complex challenges, believed to be insurmountable, will be overcome with new developments in powerful synthetic biomedical and genetic engineering technologies [29].

### 3.7. Optro

Optro was developed jointly by Somatogen Inc. in San Diego, CA, USA and Eli Lilyin Indianapolis, IN, USA. Optro is a genetically engineered, recombinantly modified hemoglobin and cross-linked with glycine. Optro is a derivative manufactured by using recombinant DNA technologies in which modified human hemoglobin genes were expressed in non-human organisms, such as *E. coli* and *yeast* [5,30]. Some amino acid sequences of natural human hemoglobin genes are modified to minimize disassociation of hemoglobin molecules into dimers in blood circulation and maintain their oxygen affinity when applying recombinant DNA technologies. The hemoglobin genes are introduced into *E. coli* cells via a plasmid vector, then the gene expression of these transferred human hemoglobin genes will induce hemoglobin protein production inside the *E. Coli*. This technique does not seem to be inexpensive; however, as a matter of fact, the high cost is one of the major hurdles for massive hemoglobin production. Optro is the first product in this category [30]. Future studies will be aimed at choosing various combinations of multiple mutations which would reduce nitric oxide scavenging, oxidative degradation, and denaturation without compromising O_2_ delivery [30]. Optro had a half-life of 2–19 h. Though reasonably tolerated by patients, Optro did not meet FDA criteria, and it was terminated in 1999 due to nitric oxide scavenging.

### 3.8. Hemospan

Hemospan was developed by Sangart Inc. in San Diego, CA, USA. Hemospan (MP-4) is a polyethylene-glycol (PEG)-conjugated human hemoglobin with increased molecular size and improved oxygen affinity properties. Preliminary clinical investigations yielded some promising results, and Hemospan did not seem to cause severe vasoconstriction as other HBOCs did. This might be explained by the fact that Hemospan is a formulation of hemoglobin molecules that are conjugated to the soluble polymer PEG. Due to this high level of water hydration of PEG solutions, the resulting conjugated hemoglobin molecules have a much higher molecular radius, rendering the molecules unable to cross intercellular junctions. Thus, Hemospan seems to have significantly less nitric oxide scavenging effects and less vasoconstriction and hypertension. Unfortunately, Hemospan did not seem to meet clinical adequacy as a blood substitute agent in Phase III clinical trials, and it was terminated in 2015 [31]. As a new derivative of PEG-modified human HBOC, Hemospan has been well tolerated by volunteers and patients in Phase I and II clinical trials, no serious adverse effects have been attributable to Hemospan, and there has been no indication of hypertensive or gastrointestinal side effects. Two Phase III trials, one conducted in Europe and one in the USA, compared Hemospan with Voluven for the effects on intraoperative volume expansion and hypotension; these trials did not yield anticipated results. We believe the two clinical trials were not well designed. Hemospan is not developed mainly as a volume expander but as an oxygen therapeutic agent [31].

### 3.9. Sanquinate

Sanguinate was developed by Prolong Inc. in South Plainfield, NJ, USA. Sanquinate is a unique PEG-modified form of bovine hemoglobin that represents the latest generation of hemoglobin-based oxygen carriers. Hemoglobin PEGylation under aerobic conditions alters its structural and functional properties significantly, promotes tetramer dissociation, increases O_2_ affinity, and abolishes O_2_ binding cooperativity [32]. However, PEGylation under anaerobic conditions with the same chemistry will lead to a low-affinity HBOC, which maintains cooperativity in O_2_ binding, and it did not cause profound hypertension in the animal model and was 99% retained in the plasma within 6 hours. When extremely low O_2_ tensions exist in ischemic tissues, the HBOC may need high affinity to optimize O_2_ delivery [32]. Sanguinate’s structural characteristics and altered hemoglobin-oxygen binding affinity allow it to bypass obstructions in the microcirculation and effectively deliver oxygen to ischemic tissues [33]. Sanguinate is also able to endogenously deliver carbon monoxide, which has been shown to reduce inflammation and oxidative stress, mitigate ischemia-reperfusion injury, and promote vasodilation. Only results from Phase I trials using Sanguinate have been published, although several Phase II trials have been completed. While these trials seemed to suggest a possible risk of myocardial injury, there was little evidence that it was due to Sanguinate. Additional larger studies are needed to better define this causal relationship. It has been successfully used under emergency circumstances in over 100 patients with severe anemia and impaired oxygen delivery where blood transfusion was contraindicated. The additional therapeutic effects (anti-inflammatory, anti-vasoconstrictive, plasma expansion) may make Sanguinate useful in the treatment of disorders in which blood is ineffective, such as stroke, inflammatory diseases, and sepsis. As such, Sanguinate is more effectively termed a resuscitation fluid [33]. Cooper et al. recently engineered a hemoglobin molecule with a single reactive cysteine residue on the surface of the α-subunit, creating a single PEGylation site (βCys93Ala/αAla19Cys). When maleimide-PEG adducts were compared with the mono-sulfone-PEG (underwent reaction at αAla19Cys), the mono-sulfone-PEG adduct seemed to be significantly more stable when incubated at 37 °C for seven days in the presence of 1 mM reduced glutathione. The authors believed that maleimide-PEGylation seemed stable enough for acute oxygen therapeutic use, but for longer vascular retention, mono-sulfone-PEG may be more appropriate [34].

### 3.10. HemoxiMer

Hemoximer was developed by Apex Bioscience in North Carolina, USA. It is a chemically modified human hemoglobin and a pyridoxalated hemoglobin polyoxyethylene (PHP) conjugate, that is why HemoxiMer is sometimes termed as PHP/HemoxiMer. Hemoximer was developed as a hemoglobin-based oxygen therapeutic but was later repurposed to be used as a scavenger of nitric oxide in any pathologic conditions with high nitric oxide. Phase I and II clinical trials on Hemoximer were completed. Two Phase III trials in these clinical settings, one in the USA and one in the EU, were partially completed. Because of the Phase III trial failure, Hemoximer development was discontinued. However, clinical investigations showed that Hemoximer was active as a nitric oxide scavenger [35].

### 3.11. OxiVita

The OxyVita hemoglobin is a new generation polymeric HBOC derived from bovine hemoglobin. It was initially invented at the University of Maryland in 1999 and further developed by OXYVITA Inc. in Middletown, NY, USA. OxyVita was manufactured by using a proprietary methodology which utilizes the zero-link polymerization technology. The process, rather than using linking agents, utilizes activators, which in turn create a super-polymer of tetramers of hemoglobin. OxyVita is a super-polymer sized at approximately 17 MegaDaltons. It was intended for the improvement of oxygen delivery when RBC transfusion was needed but either unavailable or unwanted. It was intended to overcome the issues of the past failed HBOCs [35]. OxyVita has been tested by several teams of independent researchers at various institutions over the past two decades [36]. Some clinical research projects are still actively ongoing.

## 4. Summary

The pursuit for a blood substitute thus far has been very disappointing. However, the sunrise appears to be on the horizon. The conceptual changes in recent years might have facilitated the changes in the field of blood substitutes, or oxygen therapeutics, as currently termed. It is almost impossible to artificially manufacture whole human blood, at least in the foreseeable future. Finding an alternative to its most important function, oxygen transportation, seems to be more practical and more appropriate, conceptually. Instead of looking for an agent comparable to RBC transfusion, now we are seeking for decreasing mortality in severely anemic patients due to devastating trauma, massive surgical blood loss, or other pathological condition. We discussed most HBOCs products which were either discontinued due to the FDA’s disapproval or products still under active clinical investigations. Among Hemoglobin-based oxygen delivery products, HemAssist, PolyHeme, Hemolink, Hemospan, and Hemoximer were discontinued. Hemopure is approved for clinical use in Russia and South Africa, and its sister product Oxyglobin (same as Hemopure but different trade name) has been approved for veterinary use both in the USA and the European Union [5]. HemO2life has recently been approved for organ preservation during organ transplantation in the European Union [private communication], and the developers are trying to obtain approval of this product for severe anemic conditions. OxyVita and Sanguinate are still under active clinical investigations. Hopefully the coming 10–20 years will usher in an explosive growth in oxygen therapeutics.

## Figures and Tables

**Table 1 medicina-59-00396-t001:** Hemoglobin-based oxygen carriers: a summary of HBOC products.

Hb Modifications Mechanism(s)		Product (Company)	Prominent Side Effects in Animal &/or Human Trials	Current Status
Conjugated	Maleimide-PEGylated human Hb(oxy)	Hemospan (MP40x) (Sangart, San Diego, CA, USA)	HTN, MI, CVA, TIA, ARF, high mortality	Phase II & III completed. ↑ O_2_ level, Development shelved in 2015.
	PEGylated carboxy-Hb (Bovine Hb)	Sanguinate (Prolong, South Plainfield, NJ, USA)	Dizziness, lethargy, musculoskeletal adverse events	Phase II complete, Phase III will not complete. Terminated. Research ongoing
	Conjugated PHP & Crosslinked human Hb	PHP/Hemoximer (Apex bioscience)	HTN, ARF, CVA, MI, ↑mortality	PHP: Phase II completed; Phase III terminated 2011
Crosslinked				
	α-α crosslinked (Fumarate) (human Hb)	HemAssist (Baxter, IL, USA)	HTN, MI, CVA, Higher mortality	Phase III halted in 1999
	Crosslinked & polymerized (Human Hb)	Hemolink (o-raffinose) (Hemosol, Toronto, Canada)	HTN, severe cardiotoxicity, MI, CVA, TIA, high mortality	Phase III completed. Abandoned
Polymerized				
		PolyHeme (Northfield Labs, Evanston, IL, USA)	HTN, MI, CVA, TIA, ARF, higher mortality	Phase III completed in USA, but no FDA approval. Abandoned in 2009
	Glutaraldehyde polymerization (bovine Hb)	Hemopure (Biopure)	HTN, elevated liver enzyme, methemoglobinemia, oliguria	Phase III completed & approved for human use in South Africa & Russian.
		Oxyglobin (Biopure)		Approved for animal use in USA & Europe
	Polymerized (Zero-linked) (Bovine Hb)	OxyVita (OXYVITA Inc. Windsor, NY, USA)	Animal study ongoing	Preclinical trial ongoing Research ongoing
Encapsulated	LEAcHb	Dr. Chang in McGill U		Research ongoing
	hemoglobin vesicles	Dr. Sakai’s study		Research ongoing
Natural Hb	Erythrocruorin	From earthworm Lumbricus terrestris		Research ongoing
	Polychaete Arenicola marina	HemO2Life by Hemarina, Brittany, France		Approved for clinical use by the European Union for donor organ preservation
Genetically engineered Hb	Recombinant DNA technology	Optro by Somatogen & Eli Lily	Seemed tolerated by patient, significant NO scavenging	Discontinued in 1999

HTN: hypertension, MI: Myocardial infarction, CVA: cerebrovascular accident (stroke), TIA: transient ischemic attack, ARF: acute renal failure, RBC: red blood cell, ICH: intracerebral hemorrhage, HR: heart rate, BP: blood pressure, LEAcHb: liposome-encapsulated actin-hemoglobin, NO: nitric oxide, PEG: Polyethylene Glycol. ↑ = increase

## Data Availability

Upon request, all literature from PubMed/Medline.

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
