# Peer review of "Hemoglobin-Based Oxygen Carriers: Where Are We Now in 2023?"

_medicina, 2023, doi:10.3390/medicina59020396_

Round 1
Reviewer 1 Report
The manuscript by Chen et al reports on the studies carried out on the development of blood substitutes, or more precisely on hemoglobin-based oxygen carriers.
The review is informative on the investigations, results and failures of the many products, none of them have been approved by FDA or EMEA. However, there are many weaknesses here listed:
i) The scientific level of the review is rather low, aiming more to phD students than to general readers of a high IF scientific journals, with no deep insights on issues that have so far prevented regulatory agencies approval. So at the end, readers do not really have directions and lines of evolution of the next generation of HBOCs.
ii) Authors failed to quote the books, containing many chapters, that have been published in the last 20 years by leading scientists working in this field, as well as recent reviews on the same topics. The result is a “light” story lacking the depth required by a review in this complex and challenging field.
iii) Authors failed to quote the more recent works carried out by Cooper group aimed at exploiting a combination of genetic and chemical approaches for the development of safe HBOCs.
iv) Authors failed to report any data on the oxygen binding properties of described HBOCs to give some information on the differences among products, that might correlate with efficiency and safety.
v) Authors failed to correctly indicate the role of DPG as allosteric effectors, erroneously indicated as an enzyme.
vi) Authors failed to give the right relevance to the strategy of conjugation with PEG, that has been one of the main lines of investigations, more than the conjugation with enzymes. Authors do not even mention why PEG is used.
Minor point:
E. Coli read E. coli
Many sentences end with a comma whereas a full point should be used.
Author Response
Responses to the reviewer’s comments:
The review is informative on the investigations, results and failures of the many products, none of them have been approved by FDA or EMEA.
Thank you for the comments! You are truly an expert in the field, we can tell from your questions below. Honored to have you reviewing this manuscript!
However, there are many weaknesses here listed:
- The scientific level of the review is rather low, aiming more to phD students than to general readers of a high IF scientific journals, with no deep insights on issues that have so far prevented regulatory agencies approval. So at the end, readers do not really have directions and lines of evolution of the next generation of HBOCs.
Well, this is a general review (not comprehensive review) of the current status of the field of blood substitutes/oxygen carriers. It is very true it has been very disappointing, only a few achieved limited FDA/EMEA approval.
We did not intend to go very deep into each product because we believed most readers will not be very interested in the very details of each product, and potentially more confused by the complex nature of these products.
We just want the readers to know where we are now.
- Authors failed to quote the books, containing many chapters, that have been published in the last 20 years by leading scientists working in this field, as well as recent reviews on the same topics. The result is a “light” story lacking the depth required by a review in this complex and challenging field.
We quoted significant amount of the newest book published several months ago by Liu, Kaye and Jahr by Springer Nature. Majority of the authors of the book are the leaders of the field. The book has very detailed information for those who are either working on this field or very interested in these products.
- Authors failed to quote the more recent works carried out by Cooper group aimed at exploiting a combination of genetic and chemical approaches for the development of safe HBOCs.
Thanks for mentioning this article, we added to our manuscript and references.
- Authors failed to report any data on the oxygen binding properties of described HBOCs to give some information on the differences among products, that might correlate with efficiency and safety.
That is what we are working on another paper, and potentially point out the future directions. Thank you pointing out this, this is a very important question.
- Authors failed to correctly indicate the role of DPG as allosteric effectors, erroneously indicated as an enzyme.
Corrected as suggested, thanks.
- vi) Authors failed to give the right relevance to the strategy of conjugation with PEG, that has been one of the main lines of investigations, more than the conjugation with enzymes. Authors do not even mention why PEG is used.
We only intended to give readers an overall view of the current status of the field of blood substitutes/oxygen therapeutics. There are many things we did not discuss and did not mention. Most clinicians have little understanding of the blood substitutes, we only intended to offer an overall view. Thanks.
Minor point:
- Coliread E. coli changed as suggested, thanks.
Many sentences end with a comma whereas a full point should be used.
We made some changes. Thanks.

Reviewer 2 Report
General comments:
This mini review, which gives us a summarized history of Hemoglobin-based oxygen carriers (HBOC), either as developed and then discontinued products or products still undergoing clinical research, seems to be one more attempt to fuel the need for HBOC alternatives, at least its most important function, oxygen transportation because: erythrocytes may not be available for trauma patients, patients may decline transfusion based on personal, cultural, or religious reasons, and blood product transfusions may not be necessary in some cases. It is easy to read, though the authors could have been “a bit more generous” with description and more precise when discussing some of the prominent HBOC products, after all this is a review.
Certain corrections are still needed:
Line 3. 1Lin Chen, MD, PhD, 2Zeyong Yang, MD, PhD, 3Henry Liu and MD ?
Line 33. “There are numerous etiologies to cause human beings to lose blood or not manufacturing adequate blood”. e.g., There are numerous etiologies to cause blood loss in humans or the production of normal blood cells is inadequate.
Line 146. “complicated products” should be complex products.
Line 153 and 154. Requires formatting.
Line 159. “A phase I clinical trial in humans was approved in Japan recently in November, 2022”. A phase I clinical trial in humans was approved in Japan in November 2022.
Line 163-4 Requires reconstruction e.g., “Basically, the modified human hemoglobin genes are introduced via plasmid into E. Coli or Yeast, the modified human Hb genes then expressed inside E. Coli”. Please use some parts from OPTO (7) passage to reconstruct the sentence.
Line 165-6. “Large quantity of recombinant modified human hemoglobin can also be produced and purified from transgenic E. coli. No need for “recombinant modified” construction, just recombinant would be fine.
Line 167. “to address specific efficacy”, what specific efficacy?
In the same paragraph (E) Genetically-engineered recombinant hemoglobin, line 168, I would like to know who is “WE”? Please change Again, it is more precisely written in the (7) Optro paragraph (line 309-313), since you emphasize that the product was not approved, even terminated in 1999.
Line 171. “This technique generally does not have much concerns pertinent to transmitting infectious diseases”. Please rewrite the sentence.
Line 188-9. “Here we will discuss some of the most representative products which either”. …which were either developed in the past and discontinued or they are still under clinical trials.
When describing the representative products in section III, please add the type of Hb molecular modification, it does not hurt to repeat important things (A,B,C, or combined either cross-linked and polymerized, or cross-linked and conjugated).
Line 193. “in parturient”, in patients.
Line 210. “noninferiority”, probably non-inferiority
Hemopure by Biopure is glutaraldehyde polymerized bovine Hb, and Hemolink by Hemosol is raffinose cross-linked and polymerized human Hb which is presented in Table 1 but not in the text (Section III/ 4. Hemopure, line 225-6). Further, “Hemopure is chemically stabilized, cross-linked bovine hemoglobin dissolved in a salt solution…”? Developing Hb is certainly not my line of work, but I would like to know whether there is a difference between those two in the mechanisms of Hb molecular modifications? Please compare one more time the text with Table 1
Note: Table 1 requires more attention, precision, consistency because it represents the data in the manuscript. e.g., Dr Chang and Dr Sakai, phase I clinical trial granted or approved… e.g., HemO2Life add the approval for organ transplantation… if you write high or higher why the sign ↑ in two places ( it should look uniform not sometimes this sometimes that)… e.g. Hemopure again, the authors are not talking about the side effects in the text but they appear in the table… etc.
Line 257. “HemO2life is also great for organ preservation before transplantation as a tool of preserving donor organ function in preclinical and clinical settings, that is what European Union approved for currently.” Please revise, this is a scientific paper, the authors cannot write how something “is great” for organ transplantation!!!
Line 267-269. “Hopefully in the near future that HemO2life will obtain the approval for anemic situations in the European Union or even other countries”. Should be at least like the following sentence: Hopefully in the near future HemO2life will obtain the approval for anemic conditions in the European Union or even other countries.
Line 332. “trails” should be trials.
Line 343. “ have been published”. Do we know where it was published?
Line 375. “Active clinical research activities are still actively ongoing”. Please reconstruct. e.g., Ongoing clinical research studies.
Line 384-5. “And instead of looking an agent comparable to RBC transfusion in improving patient’s overall conditions, seeking for improving mortality in”… Please reconstruct the sentence, pay attention to grammar, and I hope it would reduce the rate of mortality instead of improving.
Author Response
This mini review, which gives us a summarized history of Hemoglobin-based oxygen carriers (HBOC), either as developed and then discontinued products or products still undergoing clinical research, seems to be one more attempt to fuel the need for HBOC alternatives, at least its most important function, oxygen transportation because: erythrocytes may not be available for trauma patients, patients may decline transfusion based on personal, cultural, or religious reasons, and blood product transfusions may not be necessary in some cases. It is easy to read, though the authors could have been “a bit more generous” with description and more precise when discussing some of the prominent HBOC products, after all this is a review.
Thanks for the comments! Truly appreciated!
Certain corrections are still needed:
Line 3. 1Lin Chen, MD, PhD, 2Zeyong Yang, MD, PhD, 3Henry Liu and MD ?
Corrected as suggested
Line 33. “There are numerous etiologies to cause human beings to lose blood or not manufacturing adequate blood”. e.g., There are numerous etiologies to cause blood loss in humans or the production of normal blood cells is inadequate.
Changed as suggested.
Line 146. “complicated products” should be complex products.
Changed as suggested.
Line 153 and 154. Requires formatting.
The journal editors will work on it. Thanks. We noticed that also.
Line 159. “A phase I clinical trial in humans was approved in Japan recently in November, 2022”. A phase I clinical trial in humans was approved in Japan in November 2022.
Changed as suggested.
Line 163-4 Requires reconstruction e.g., “Basically, the modified human hemoglobin genes are introduced via plasmid into E. Coli or Yeast, the modified human Hb genes then expressed inside E. Coli”. Please use some parts from OPTO (7) passage to reconstruct the sentence.
Changed as suggested.
Line 165-6. “Large quantity of recombinant modified human hemoglobin can also be produced and purified from transgenic E. coli. No need for “recombinant modified” construction, just recombinant would be fine.
Changed as suggested.
Line 167. “to address specific efficacy”, what specific efficacy?
“oxygen-carrying” efficacy, changed as suggested.
In the same paragraph (E) Genetically-engineered recombinant hemoglobin, line 168, I would like to know who is “WE”? Please change Again, it is more precisely written in the (7) Optro paragraph (line 309-313), since you emphasize that the product was not approved, even terminated in 1999.
Haha, “We” means scientists or whoever is working on this field. Thanks.
Line 171. “This technique generally does not have much concerns pertinent to transmitting infectious diseases”. Please rewrite the sentence.
Modified the sentence as suggested.
Line 188-9. “Here we will discuss some of the most representative products which either”. …which were either developed in the past and discontinued or they are still under clinical trials.
When describing the representative products in section III, please add the type of Hb molecular modification, it does not hurt to repeat important things (A,B,C, or combined either cross-linked and polymerized, or cross-linked and conjugated).
Yes, you are right, thanks.
Line 193. “in parturient”, in patients.
We believe using parturient is better because it was used in the original paper. Thanks.
Line 210. “noninferiority”, probably non-inferiority
Hemopure by Biopure is glutaraldehyde polymerized bovine Hb, and Hemolink by Hemosol is raffinose cross-linked and polymerized human Hb which is presented in Table 1 but not in the text (Section III/ 4. Hemopure, line 225-6). Further, “Hemopure is chemically stabilized, cross-linked bovine hemoglobin dissolved in a salt solution…”? Developing Hb is certainly not my line of work, but I would like to know whether there is a difference between those two in the mechanisms of Hb molecular modifications? Please compare one more time the text with Table 1
Note: Table 1 requires more attention, precision, consistency because it represents the data in the manuscript. e.g., Dr Chang and Dr Sakai, phase I clinical trial granted or approved… e.g., HemO2Life add the approval for organ transplantation… if you write high or higher why the sign ↑ in two places (it should look uniform not sometimes this sometimes that) e.g. Hemopure again, the authors are not talking about the side effects in the text but they appear in the table… etc.
You are correct. Dr. Chang and Dr. Sakai don’t have a product’s commercial name. So we used their names instead.
Pertinent changes have been made as suggested.
Line 257. “HemO2life is also great for organ preservation before transplantation as a tool of preserving donor organ function in preclinical and clinical settings, that is what European Union approved for currently.” Please revise, this is a scientific paper, the authors cannot write how something “is great” for organ transplantation!!!
Changed to “indicated”, thanks!!
Line 267-269. “Hopefully in the near future that HemO2life will obtain the approval for anemic situations in the European Union or even other countries”. Should be at least like the following sentence: Hopefully in the near future HemO2life will obtain the approval for anemic conditions in the European Union or even other countries.
Changes made as suggested.
Line 332. “trails” should be trials.
Changed as suggested, thanks.
Line 343. “ have been published”. Do we know where it was published?
In the ref 32.
Line 375. “Active clinical research activities are still actively ongoing”. Please reconstruct. e.g., Ongoing clinical research studies.
Modified as suggested. Thanks!
Line 384-5. “And instead of looking an agent comparable to RBC transfusion in improving patient’s overall conditions, seeking for improving mortality in”… Please reconstruct the sentence, pay attention to grammar, and I hope it would reduce the rate of mortality instead of improving.
Changed as suggested. Thank you so much.

Round 2
Reviewer 1 Report
Authors did not significantly improved the manuscript, as some of the comments were not at all considered, such as indication of oxygen binding properties and PEG-based HBOCs. Overall, I feel that the review does not provide the level of information required by the complex field of HBOC.
Author Response
Responses to reviewer’s comments
Thanks for the reviewer’s thoroughness!!!
Authors did not significantly improved the manuscript, as some of the comments were not at all considered, such as indication of oxygen binding properties and PEG-based HBOCs. We added a small portion to “9. Sanguinate” to show the effect of PEGylation on O2 binding.
This article is intended to give the readers the current product status, and inform them that breakthroughs are on the horizon!!! This article is NOT intended to explain everything related to HBOC.
If you like to know more, please read the book:
https://www.amazon.com/Blood-Substitutes-Oxygen-Biotherapeutics-Henry-ebook/dp/B0B8YCL3KL/ref=sr_1_1?crid=2W8FPPHX536UI&keywords=blood+substitutes+Liu&qid=1675395908&sprefix=blood+substitutes+liu%2Caps%2C178&sr=8-1
This book’s authors are mostly the experts in pertinent products.
Overall, I feel that the review does not provide the level of information required by the complex field of HBOC.
We have to respectively disagree! This relative short review gives readers the current status of HBOCs. Table 1 in this article you will not see any where else.
